# The impact of data resolution on dynamic causal inference in multiscale ecological networks
Erik Saberski [1] ✉, Tom Lorimer[1,2,3], Delia Carpenter[1], Ethan Deyle [4], Ewa Merz [1,5], Joseph Park [1,6], Gerald M. Pao[6] & George Sugihara[1] ✉

While it is commonly accepted that ecosystem dynamics are nonlinear, what is often not acknowledged is that nonlinearity implies scale-dependence. With the increasing availability of high-resolution ecological time series, there is a growing need to understand how scale and resolution in the data affect the construction and interpretation of causal networks—specifically, networks mapping how changes in one variable drive changes in others as part of a shared dynamic system ("dynamic causation"). We use Convergent Cross Mapping (CCM), a method specifically designed to measure dynamic causation, to study the effects of varying temporal and taxonomic/functional resolution in data when constructing ecological causal networks. As the system is viewed at different scales relationships will appear and disappear. The relationship between data resolution and interaction presence is not random: the temporal scale at which a relationship is uncovered identifies a biologically relevant scale that drives changes in population abundance. Further, causal relationships between taxonomic aggregates (low-resolution) are shown to be influenced by the number of interactions between their component species (high-resolution). Because no single level of resolution captures all the causal links in a system, a more complete understanding requires multiple levels when constructing causal networks.

One of the fundamental goals of ecology is to understand causal interactions as they occur within naturally evolving ecosystems. Here causation can be direct or transitive, span multiple mechanisms (e.g., trophic, competition, mutualism, etc.), and change with ecosystem state. All of this ultimately determines how effects (natural or managed) propagate, and travel in ways that can sometimes lead to unintended consequences. Although controlled experiments can be important for establishing direct causal links *in principle*, in practice, because interactions in nature tend to change with the evolving ecosystem state[1–4] static single-factor assessments fail to translate into predictive understanding. This is a challenge that can be met with a data-driven approach for inferring causal effects between ecosystem components using observational time series[4–10].

How data resolution impacts perception is fundamental. Indeed, the basic notion of what constitutes the variables of study in real ecological applications is necessarily tied up in the scales of observation. For example, in some lakes we might measure chlorophyll-a every hour but only at the surface, while in others we might measure and define each known species of chlorophyte and diatom at various depths, but only once per summer. Although both observe something of dynamics underlying primary production, such differences in scale and aggregation will determine what we see as the causal factors shaping the dynamics of those observations. Accounting for these differences is the focus of this work.

While approaches that rely on a statistical framework do not assume an underlying deterministic dynamical system[11–13], here we take the position that dynamics are an essential part of the machinery. Through a dynamical systems lens, causality can be regarded as explicitly deterministic, mechanistic, and dynamic. This contrasts with statistical definitions of causality where relationships are independent of changing system states. Thus, we are interested in whether a change in one variable produces a

[1]Scripps Institution of Oceanography, 9500 Gilman Drive, La Jolla, CA, 92093-0206, USA. [2]Eawag, Swiss Federal Institute of Aquatic Science and Technology, Surface Waters – Research and Management, Kastanienbaum, Switzerland. [3]Stream Ocean AG, Zurich, Switzerland. [4]Department of Biology, Boston University, Boston, MA, 02215, USA. [5]Eawag, Swiss Federal Institute of Aquatic Science and Technology, Department of Aquatic Ecology, Duebendorf, Switzerland. [6]Okinawa Institute of Science and Technology Graduate University, Biological Nonlinear Dynamics Data Science Unit, 1919-1 Tancha, Onna-son, Okinawa, 904-0495, Japan. ✉e-mail: eriksaberski@gmail.com; gsugihara@gmail.com

change in another due to their mechanistic coupling in a shared dynamic system (i.e., "dynamic causation").

Here, we revisit the role of scale and aggregation in causal pattern and process using a common data-driven approach specifically aimed at measuring dynamic causation in ecosystems: convergent cross-mapping (CCM)[4,8–10,14,15]. CCM infers causal relationships from time series data by exploiting Takens' Theorem[16], which states as a generic property that, quite remarkably, any one variable in a coupled dynamic system will contain information about the other variables in the network. This means that links inferred using CCM are not simply direct, binary and independent, but include transitive effects across multiple components of the full natural system. Thus, causal interaction webs produced by CCM provide a comprehensive picture of causal interdependence that can be used, for example, to effectively study direct and indirect consequences of interventions, and, in principle, it should be able to do so using readily available monitoring data.

Unlike classical structural modeling approaches for detecting causal association[11–13], CCM is specifically designed to detect nonlinear relationships that are invisible to correlation-based methods. Adding to its practical significance in ecological network analysis is the fact that CCM does not require all relevant causal variables to be observed—a consequence of Takens' Theorem. Other commonly used methods to construct causal networks from time series data include Granger causality and structural causal models informed by machine learning[17]. None of these methods are both explicitly nonlinear and dynamic, and many are based on conditional probabilities that require all relevant causal variables to be observed. This is a constraint that makes them less practical for ecological applications. It is worth noting that when it is not possible to measure all system variables, focusing on the dominant variables and treating the others as noise can be beneficial. Further, these methods focus on direct linkages which allows for easier conceptualization, yet precisely for that reason they do not capture the true level of interdependence in ecosystems. CCM can address the kinds of problems that are directly relevant to conservation and management efforts[18] —for example, how small changes in one variable can propagate and push a system toward or pull it away from collapse[19,20].

Convergent Cross-Mapping (CCM) has its limitations, particularly in the context of high-dimensional chaotic systems where finding good analogues, as required by Takens' Theorem, can be challenging. Work by Bradley and Kantz[21], Ruelle[22], and Baldovin, Cecconi, and Vulpiani[23] highlight the difficulties of time series reconstruction in such high-dimensional contexts. Although Takens' theorem was originally formulated for deterministic systems, extensions by Stark et al.[24] have expanded its applicability to certain classes of stochastic dynamical systems. Further, the simplex and S-map algorithms utilized in our empirical dynamic modeling (EDM) analyses are specifically designed to handle observation noise, relying on the existence of low-dimensional deterministic relationships between variables (e.g., Sugihara et al. (2011)[25]). These extensions ensure that our embedding procedures remain valid and robust, even in the presence of stochastic elements, thus supporting the applicability of CCM to real-world measurements.

Indeed, whichever causal inference method is chosen, it is an unavoidable fact that data will be aggregated through primary observations and subsequent processing over some spatial, taxonomic, and/or temporal scale in constructing any kind of ecological network. Food webs are often constructed in terms of functional groups by pooling species into taxonomic aggregates[26] as well as trophic equivalence classes[27] and pollination networks have been analyzed at spatial and temporal scales that span many orders of magnitude[28,29]. Some have argued that aggregated data can reveal robust patterns and valuable insight[30–33], while others suggest that aggregation can mask important ecosystem dynamics that arise more coherently at finer scales[34–36]. Taking in both points of view suggests that one can be intentional about the often-unacknowledged choice of scale and even take full advantage of it to improve understanding of ecological dynamics on scales relevant to management.

As early as 1992, in his MacArthur Award keynote address Simon Levin stated: "Applied challenges, such as the prediction of the ecological causes and consequences of global climate change, require interfacing phenomena that occur on very different scales of space, time, and ecological organization"[37]. Indeed, the recognition by ecologists[38–40] that dynamic processes occur simultaneously at multiple spatial and temporal scales has arisen in many fields including economics (e.g., Lange-Hicks Condition[41]) and neuroscience[42], where it is often termed the "aggregation problem". Here the focus has been on investigating conditions under which a coarse-grained "macrosystem" view (where dynamics occur between aggregated macroscopic variables like functional groups) and a fine-grained "microsystem" view (where dynamics occur between disaggregated or less aggregated variables like population abundances of individual species) give different results[31,38]. The simple answer is that unless the dynamics are linear or can be separated (e.g., where fast components can be treated as if they are in equilibrium and slow components as if they are constant, c.f. Tychonov's dimension reduction[38]), scale matters. Indeed, the overwhelming evidence that ecological dynamics are nonlinear and state-dependent means that analyses at different scales will present different portraits of the functional relationships – a potential liability if ignored, but when accounted for can become a substantial asset to support the understanding and management of these systems.

Because nonlinearity implies scale-dependence and ecological dynamics are nonlinear[43–46], it is not surprising that identifying dynamic causal linkages will necessarily depend on the scale and resolution of the data used. However, how this plays out in practice is not known. Here we aim to provide a better understanding of the implications of scale and resolution when constructing and interpreting dynamic causal networks for ecosystems.

## Methods
### Metrics
In this study, we employ a few metrics to evaluate the structure and dynamics of ecological networks: Fine-Scale Connectance, Resolved Aggregate Interaction Strength, and Aggregated Functional Group Linkage.

**Fine-Scale Connectance** refers to the proportion of potential links between individual species that are realized in the network. It is a measure of how interconnected the species are at a detailed, species-level resolution. Higher connectance indicates a more interconnected network.

**Resolved Aggregate Interaction Strength** measures the strength of interactions between aggregated functional groups, rather than individual species. This metric is derived by summing the abundances of individual species within each group and then assessing the causal influence between these groups. It captures the net effect of multiple species interactions within and between groups, providing a more generalized view of ecosystem dynamics. This is particularly useful when studying systems at a broader scale where detailed species-level data may not be available or practical to analyze.

**Aggregated Functional Group Linkage** indicates the connections between functional groups in an aggregated network. Functional groups here are collections of species such as diatoms or benthic herbivores. This metric assesses how these groups are linked through direct and indirect interactions.

### Convergent cross mapping
Convergent Cross Mapping (CCM)[15] is a technique designed to infer causality from time series data by leveraging the principles of dynamical systems theory. It is particularly well-suited for identifying nonlinear and non-separable relationships, which are common in ecological systems. The method hinges on Takens' Theorem[16], which asserts that the state space of a dynamical system can be reconstructed from the time series of a single variable within the system. This theorem underpins the notion that each variable in a coupled dynamic system contains information about other variables, enabling the reconstruction of the entire system's dynamics from any single time series.

Consider two time series $X(t)$ and $Y(t)$ representing two variables in a coupled dynamical system. If $X$ causally influences $Y$, then states of Y can be

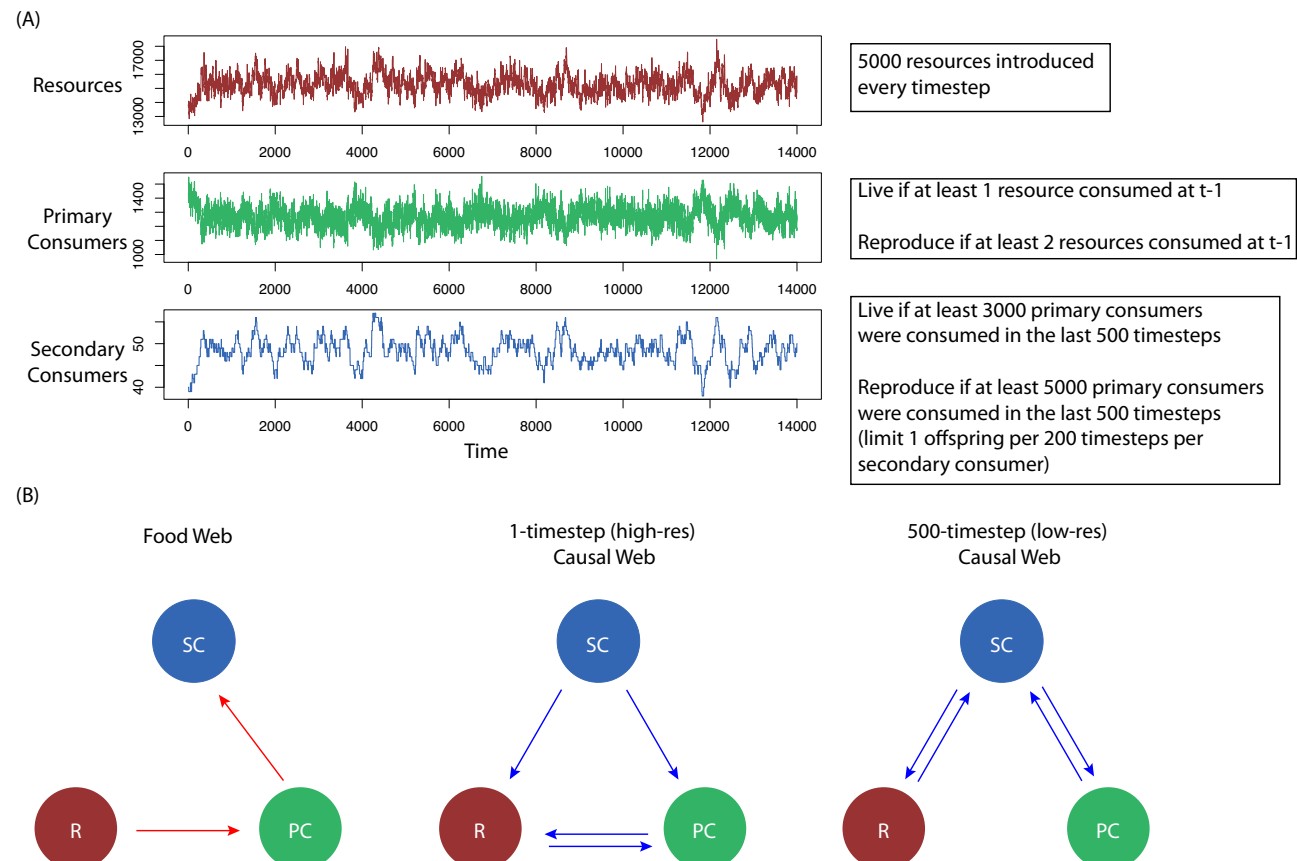

**Fig. 1 | A model simulation in which resources, primary consumers, and secondary consumers exist on a grid and move randomly. A** Timeseries generated from the model simulation. **B** Different interaction networks that represent the model. Each timestep, primary consumers eat resources and secondary consumers consume primary consumers. Both primary and secondary consumers have rules determining whether they survive, starve, or reproduce. Primary consumers and resources interact on a 1-timestep timescale, and secondary consumers consume primary consumers at a 1-timestep scale as well; however, primary consumers influence secondary consumers at a 500-timestep scale. Thus, causal webs constructed using a 1-timestep frequency do not resolve the influence of primary consumers on secondary consumers (**B**, middle), but the causal webs constructed using a 500-timestep frequency do (**B**, right).

used to predict the states of X. CCM quantifies this relationship by cross mapping the reconstructed state spaces derived from the time series data.

To apply CCM, we first reconstruct the state spaces of $X$ and $Y$ using time-delay embedding. For a time series $X(t)$, the reconstructed state space $MX(t)$ is formed by:

$$M_x(t) = \{X(t), X(t-\tau), X(t-2\tau), \cdots, X(t-(E-1)\tau)\}$$

Here, $\tau$ is the time delay, and $E$ is the embedding dimension. Similarly, the state space $MY(t)$ for $Y(t)$ is constructed in the same manner.

Once the state spaces are reconstructed, CCM measures how well the states of one variable can be predicted using the states of the other. Specifically, we use $MY(t)$ to cross map and predict $X(t)$

$$\hat{X}(t) = \sum w_i X(t_i)$$

Here, $\hat{X}(t)$ is the predicted value of $X(t)$, and $t_i$ are the indices of the nearest neighbors to Y(t) in $MY(t)$. The weights $w_i$ are determined by the distances between $Y(t)$ and its neighbors, using an exponential weighting scheme.

The strength of the causal relationship is quantified by the correlation between the predicted values $\hat{X}(t)$ and the observed values $X(t)$:

$$\rho = \text{correlation}(\hat{X}(t), X(t))$$

A high correlation $\rho$ indicates that $X$ can be reliably predicted from $Y$, suggesting that $Y$ contains information about $X$, consistent with a causal influence of $X$ on $Y$.

## Model and Field Data

To understand how dynamics spanning multiple time scales can be accommodated we construct a simple game-of-life analogue that incorporates trophic activity on multiple time scales. It is an individual-based ecological automata model with three components intended to simulate species dynamics in three trophic levels, each operating on a different time scale (Fig. 1): "resources", "primary consumers", and "secondary consumers". The components (individuals) exist in a 2-dimensional (1000 ×1000) grid. Each individual moves randomly and follows the simple trophic rules described below to determine whether it survives or reproduces in subsequent timesteps. The system is initialized arbitrarily with 15,000 resources, 1,500 primary consumers, and 20 secondary consumers placed randomly on the grid. Each is allowed to move randomly from their position in both x and y directions with a speed (distance traveled in one timestep) $s$ ($s_R$ and $s_{PC} = 10$, $s_{SC} = 25$). Primary consumers eat resources within a radius $r_{PC} = 10$ and secondary consumers consume primary consumers within a radius $r_{SC} = 100$. At each timestep, 500 new resources are introduced randomly on the grid. This pedagogical example is intended to be a simple template to demonstrate scale effects, and the qualitative results are robust to the specific parameters chosen.

If an individual primary consumer consumes at least 1 resource at time t, it survives to time t + 1; if it consumes at least 2 resources at time t it will have an offspring at time t + 1. However, if a primary consumer consumes 0 resources at time t, it does not survive to time t + 1. Secondary consumers follow similar rules but operate on much larger scales: it survives if it has consumed at least a minimum number of 3000 of primary consumers in the prior 500 timesteps and has one offspring if it consumes at least 5,000

primary consumers in the last 500 timesteps (limited to 1 offspring per 200 timesteps per secondary consumer).

The simulation ran for 15,000 timesteps to generate three distinct abundance time series, one for each trophic category (Fig. 1A). We performed CCM between each time series using $E = 1{:}10$, tau = 1, tp = -1 (high-resolution web, Fig. 1B middle) and $E = 1{:}10$, tau = 500, tp = $-500$ (low-resolution web, Fig. 1B right). Only CCM linkages having rho values greater than the maximum absolute cross correlation at any value of E were taken to show nonlinear causal connection. Subtracting out the linear cross-correlation is a simple way to measure whether there are causal dynamics beyond the linear correlation[47]. Figure S1 shows the CCM results at varying embedding dimensions.

We used a simple logistic model to examine the relationship between species-resolved connectance and resolved causal influence between two aggregated functional groups. This was accomplished by simulating 20 timeseries loosely representing 10 predators and 10 prey. A randomized interaction matrix defined the one-timestep relationships of each timeseries on each other. Predators had negative influences on prey and prey had positive influences on predators. For simplicity, the model simulations did not include any intra-aggregate interactions. The abundances were constrained by taking the reciprocal of any abundance if its value exceeded 1 at time t. Example timeseries can be found in Fig. S2. A connectance parameter (C) determined the number of non-zero elements in the interaction matrix. The main diagonal of the interaction matrix was set to -0.15 for all timeseries.

The future abundances of prey ($P$) and predators ($R$) at time $T+1$ are determined by the interaction matrices and the abundances at time $T$. The equations can be expressed as:

$$P_{T+1} = P_T + A_{PP} \cdot R_T - \alpha \cdot P_T$$

$$R_{T+1} = R_T + A_{RP} \cdot P_T - \alpha \cdot R_T$$

where:
- $P_T$ and $R_T$ are the prey and predator abundances at time $T$ respectively.
- $A_{PP}$ is the prey interaction matrix (negative influences of predators on prey).
- $A_{RP}$ is the predator interaction matrix (positive influences of prey on predators).
- $\alpha$ is the autocorrelation strength (0.15 in this model).

To constrain the abundances a piecewise function is applied as follows:

$$P_{T+1,i} = \begin{cases} \frac{1}{P_{T+1,i}} & \text{if } P_{T+1,i} > 1 \\ P_{T+1,i} & \text{if } P_{T+1,i} \leq 1 \end{cases}$$

$$R_{T+1,i} = \begin{cases} \frac{1}{R_{T+1,i}} & \text{if } R_{T+1,i} > 1 \\ R_{T+1,i} & \text{if } R_{T+1,i} \leq 1 \end{cases}$$

where:
- $P_{T+1,i}$ is the abundance of the $i$-th prey species at time $T+1$.
- $R_{T+1,j}$ is the abundance of the $j$-th predator species at time $T+1$.

We performed 500 model simulations with C ranging between 0.3 and 0.9. After each simulation completed, we took the sum of the 10 predators and 10 prey to generate two aggregate timeseries. CCM was then performed to measure the influence of the predators on prey using $E = 5$, tau = 1, and tp = -1. However, we find these results are robust to embedding dimension (Fig. S3). The resolved interaction strength was measured as the correlation coefficient between observations and predictions from this CCM analysis. This was used to explore the effect of connectance on aggregated CCM values.

We focus on four exceptional long-term ecological monitoring studies containing highly resolved time series for the individual taxa located in 1) The North Sea (from the Survey of the Marine Biological Association, formerly the Sir Alister Hardy Foundation for Ocean Science (SAHFOS), 2) Port Erin Bay (MetaBase), 3) Lake Zurich[48] and 4) A kelp forest system of San Nicholas Island[49]. Sites 1-3 were sampled monthly while the kelp forest system was analyzed as annual averages. The four studies were chosen based on data quality: their time-series data have a high degree of continuity and overall length, and there is sufficient knowledge about the ecosystems to construct a credible food web from systematic literature searches and expert knowledge. To ensure quality and uniformity in the analyses for each dataset, taxa whose time series contain less than 35 non-zero data points or were known to be inconsistently monitored were removed from the analysis.

Systematic literature surveys were used to construct food webs for each system. Data extracted for each taxon included: genus, species, prey, predators, trophic role (autotroph, heterotroph, mixotroph, primary consumer, secondary consumer), and additional ecological or biological notes of interest (including, but not limited to competition, known defenses and size). In certain cases where no information could be found for a taxon, researchers from the long-term monitoring sites were solicited to provide expert opinion to fill in remaining gaps.

Final food-web constructions represent the collation of taxon-specific relationships into functional group aggregates (Fig. 5). Functional groups are defined by both trophic level and taxonomic criteria (e.g., Omnivorous copepods). Food-web interactions were drawn between functional groups if any member within one group had a direct trophic interaction with any member within another group. Food webs for Port Erin Bay and the North Sea datasets were reviewed by plankton experts at SAHFOS.

## Analysis

Convergent cross-mapping (CCM) measures dynamic causation by using cross-map prediction to assess how well one time series can be used to predict another. If time series Y has been influenced by a driver X, it contains information that can be extracted (using Takens' Theorem[16]) to predict ("map onto") values of time series X[15]. Thus, in CCM the recipient time series has information that allows one to recover states of the driver, where predictions are based on time-indexed nearest-neighbors[50] in time-lagged embeddings. For the monthly sampled systems, we use an embedding dimension (E) of 12 and for the Kelp Forest system we use an E of 4 with the prediction horizon (tp) set to 0. A "CCM value" is defined as the Pearson's correlation between observed and predicted values.

Species abundance timeseries often exhibit spikes spanning multiple orders of magnitude. To lessen these spikes' influence on relative nearest neighbor distances in state-space reconstructions, we normalized each timeseries by subtracting the minimum value and dividing by the difference between the maximum and minimum values (bounding values between 0 and 1), then taking the square root of each value.

The abundances of species within each monthly-sampled system show high levels of seasonal synchrony. As described in Sugihara et al. (2012), when time series are synchronized, if applied uncritically CCM can return a false positive result[51,52]. To address this, "seasonal surrogates" can be constructed that maintain the seasonal relationship between two time series but shuffle the other properties of the time series. This is accomplished here by randomly shuffling the time series values within each month (e.g., shuffling all January values, then all February values, etc.). CCM is then performed on the resulting null surrogate time series to measure how accurate cross-mapping is on samples having only this seasonal property but whose dynamics are otherwise randomized. We repeat this 100 times for each possible interaction to get a distribution of null surrogate CCM values. A link is included if the CCM value is higher than at least 95 out of the 100 surrogate values.

To compare food webs and causal webs at the same aggregate resolution, we create aggregate time series by normalizing each species-abundance time series between 0 and 1 then add their abundances at each point in time. This normalization procedure gives species equal

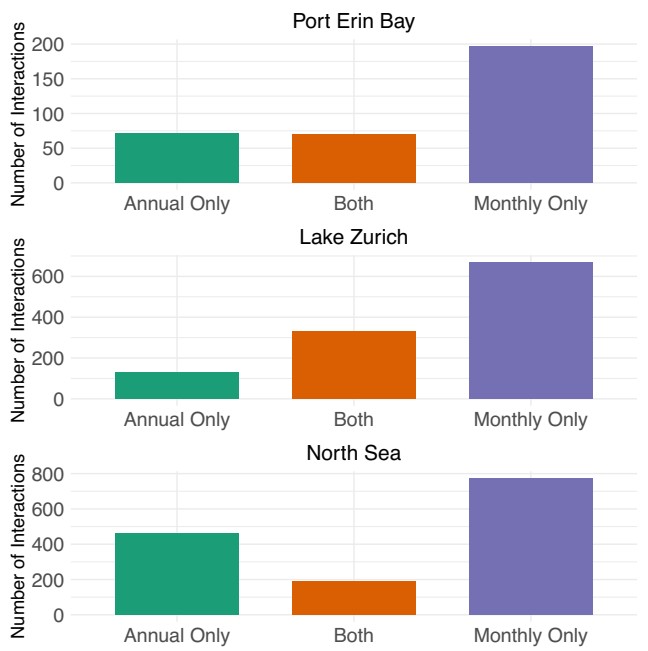

**Fig. 2 | The number of interactions in each system resolved at a monthly timescale (tau = 1) and annual timescale (tau = 12).** Note that in all systems more interactions are resolved at the monthly timescale, but there are still interactions in each system that are exclusively resolved at the annual timescale.

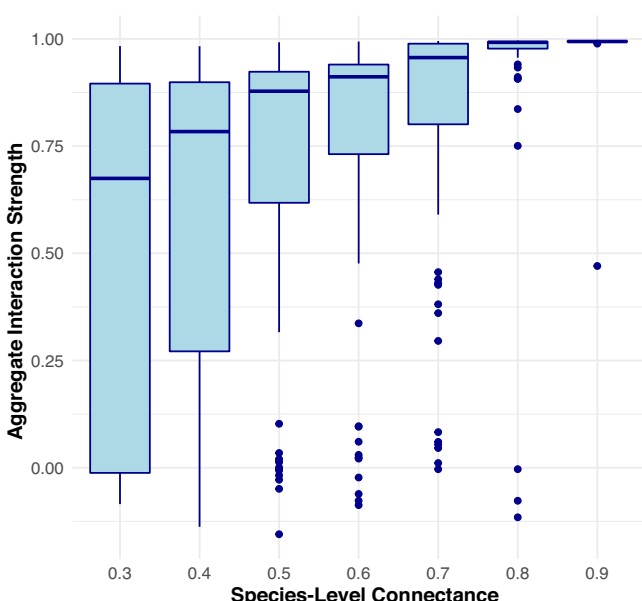

**Fig. 3 | Fine scale connectance translates to aggregate interaction strength:** Logistic models with two aggregates (10 predators and 10 prey) run at varying levels of high-resolution connectance (proportion of non-zero elements joining the two aggregates in the interaction matrix). Higher connectance at the fine scale creates stronger aggregated interaction strength resolved between the functional groups.

contribution to the aggregate time series which prevents a single, highly abundant taxa from dominating the aggregate. We then perform CCM as described above between each aggregate of each system. High-resolution webs are constructed by performing CCM between each individual species (when the system is viewed at highest resolution).

### Statistics and reproducibility

All CCM analyses were performed in the R programming language using rEDM (0.7.5). All parametric choices for CCM are described above.

### Reporting summary

Further information on research design is available in the Nature Portfolio Reporting Summary linked to this article.

### Results

First, we investigate how varying timescales can play a role in resolving ecosystem dynamics using the individual-based automata (IBA) model. By adjusting the time-lag (tau) in reconstructed embeddings to capture causal relationships at different timescales, we find that high-frequency causal webs resolve bidirectional influences between resources and primary consumers, but only show unidirectional effects of secondary consumers on primary consumers and resources (Fig. 1). The high-frequency causal web does not show any influence of primary consumers or resources on secondary consumer abundance. However, these dynamics are well-resolved at a 500-timestep scale (Fig. 1).

Similarly, we use CCM to quantify interactions between individual species at both a monthly-scale (tau = 1) and annual-scale (tau = 12) on the three monthly-sampled. This creates two distinct networks for each system: a monthly timescale network and an annual-timescale network. From these networks, interactions between species can be split into three categories: an interaction that appears in *both* networks, an interaction that only appears in the monthly-scale network, or an interaction that only appears in the annual-scale network. Fig. 2 shows the number of network interactions resolved by CCM at each time scale for each of the three categories. All three systems had more interactions resolved at the monthly scale than at the annual scale (tau = 12).

By varying the connectance of the interaction matrix of the logistic models, we find a positive association between model connectance and resolved interaction strength between the predator and prey aggregates (Fig. 3). That is, the more links there are connecting individual species across aggregates, the more likely there will be a resolved link between the aggregates in a lower-resolution network. A similar pattern is shown in real-world data (Fig. 4).

Finally, we compared the low-resolution aggregate webs with food webs constructed at the same taxonomic resolution (Fig. 5). We find that most food web links are resolved to be causal, but there are also causal interactions that are non-trophic, and trophic interactions that are not resolved as causal. The lack of a resolved causal link does not mean that interaction is absent; rather, this means that there is no detectable dynamic dependence between abundances at that scale.

### Discussion

The individual-based automata (IBA) is a great example of interactions occurring at varying timescales in a controlled system. Here, we can map out all expected interactions as a ground truth:

#### Direct interactions

- Primary Consumers (PC) → Resources (R): Primary consumers influence resource abundance by consuming them. This interaction occurs at a 1-timestep scale (PC → R).
- Resources (R) → Primary Consumers (PC): This bi-directional relationship exists because the survival and reproduction of primary consumers depend on the availability of resources. This interaction also occurs at a 1-timestep scale (R → PC).
- Secondary Consumers (SC) → Primary Consumers (PC): Secondary consumers prey on primary consumers, lowering the PC their abundance. This direct interaction occurs at a 1-timestep scale (SC → PC).
- Primary Consumers (PC) → Secondary Consumers (SC): The survival and reproduction of secondary consumers depends on the availability of primary consumers. This direct interaction occurs at a 500-timestep scale (PC → SC).

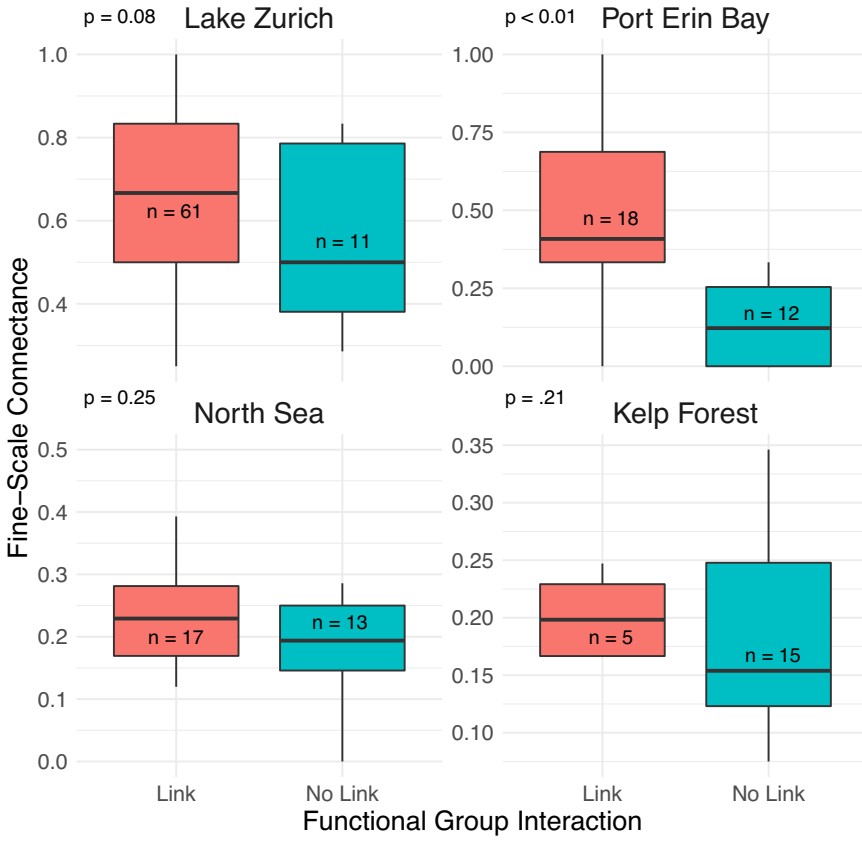

**Fig. 4 | Real world examples showing the relationship between aggregated and fine-scale linkages.** Boxplots show that causally linked aggregates (labeled 'linked') have more causal links at the species level (fine-scale), while unlinked aggregates ('unlinked') do not. Center line is the median, box limits represent upper and lower quartiles and whiskers are 1.5x interquartile range. P-values are calculated using a one-sided t-test.

## Indirect interactions

- Resources (R) → Secondary Consumers (SC): Indirectly, the availability of resources influences secondary consumers by affecting the population of primary consumers, which are prey for secondary consumers. This indirect interaction is resolved at about a 500-timestep scale (R → PC→ SC).
- Secondary Consumers (SC) → Resources (R): Indirectly, secondary consumers influence resource abundance by affecting the population of primary consumers, which consume the resources. This indirect interaction is also resolved at about a 1-timestep scale (SC → PC→ R).

Thus, there are a total of six expected interactions between three species, which implies full connectance (everything causally influences everything). However, these interactions span different timescales.

When we preform CCM on these model timeseries, we find different causal relationships appear when data of different temporal resolution is used. This is expected because, as described above, we know these interactions should exist at different timescales. Although primary consumers (and resources, indirectly) influence secondary consumer abundance, it makes sense that the causal relationship will not be resolved using high-frequency data, as the influence of primary consumers on secondary consumers occurs over 500 timesteps. When dynamics are measured at a 500-timestep scale, the influences of resources and primary consumers on secondary consumers are resolved (Fig. 1).

If this were a real system and causal linkages were only measured at a high-resolution timescale, one may inaccurately conclude that the primary consumers have no influence on the abundance of secondary consumers. In a management situation, one may then incorrectly deem it safe to alter the abundance of the primary consumers (e.g., increasing fishing, removing habitat, etc.) without any predicted consequences on the abundance of the secondary consumer. Of course, this would be a mistake since the secondary

consumers are entirely dependent on the primary consumers, but at a much slower temporal scale.

The causal influences measured depend on the scales defined by the data and by parametric decisions (e.g., embedding dimension chosen for the analysis). If the data is aggregated, the analysis will quantify relationships between aggregates. Similarly, if time series are sampled at a specific frequency (e.g., monthly), then relationships that occur at that time scale will be measured (assuming time-lagged embeddings are made with 1-timestep). Interactions occurring at higher (or lower) frequencies than those captured by the sampling frequency will not be properly resolved.

We see similar patterns in real-world data. Fig. 2 shows interactions from three distinct ecosystems, each evaluated annually and monthly. This comparison illuminates the dynamic nature of ecological interactions that manifest differently across these timescales: certain interactions become apparent only in the annual data, whereas others are evident exclusively in the monthly data. Thus, the temporal scale chosen will influence the resolved interaction network.

As a crude heuristic, it has been argued that scaling state-space-reconstruction to the generation times of species may help resolve dynamics[53]. For example, In Lake Zurich the animal species with the smallest ratio of the number of annual versus monthly resolved links were Cyclopoida C1-C3 (34 monthly, 13 annual), nauplii (32 monthly, 13 annual) and eggs (24 monthly, 10 annual) which either die or grow into their next life-stage on the order of days to weeks. The animal species with the largest ratio were adult Cyclops (9 monthly, 14 annual) and Eudiaptomus Gracilis (13 monthly, 16 annual) which has generation times on the order of months[54] (highlighted in Table S1). However, because species can exhibit dynamics that span many orders of magnitude (e.g., as illustrated in Fig. 1) it is not surprising to find exceptions to this rule-of-thumb.

Similar to temporal scale, taxonomic aggregation can also be a double-edged sword that can obscure or clarify interaction patterns. Some have argued that aggregating abundances across multiple individuals,

**Fig. 5 | A comparison of food webs and aggregate causal webs for the four systems studied.**
**A** Aggregated causal webs (blue arrows) overlayed with food webs (red arrows). **B** High-resolution causal webs mapping interactions between individual species. Each large circle represents an aggregated function group of species, and each dot represents an individual species. Note that although there are more causal links (blue arrows) than food web links (red arrows) in A, not every food web link is detectably causal as might be expected from scale considerations. Indeed, when the systems are views through a high-resolution (species-resolved) lens, there is always at least one link between all trophically linked aggregates (B).

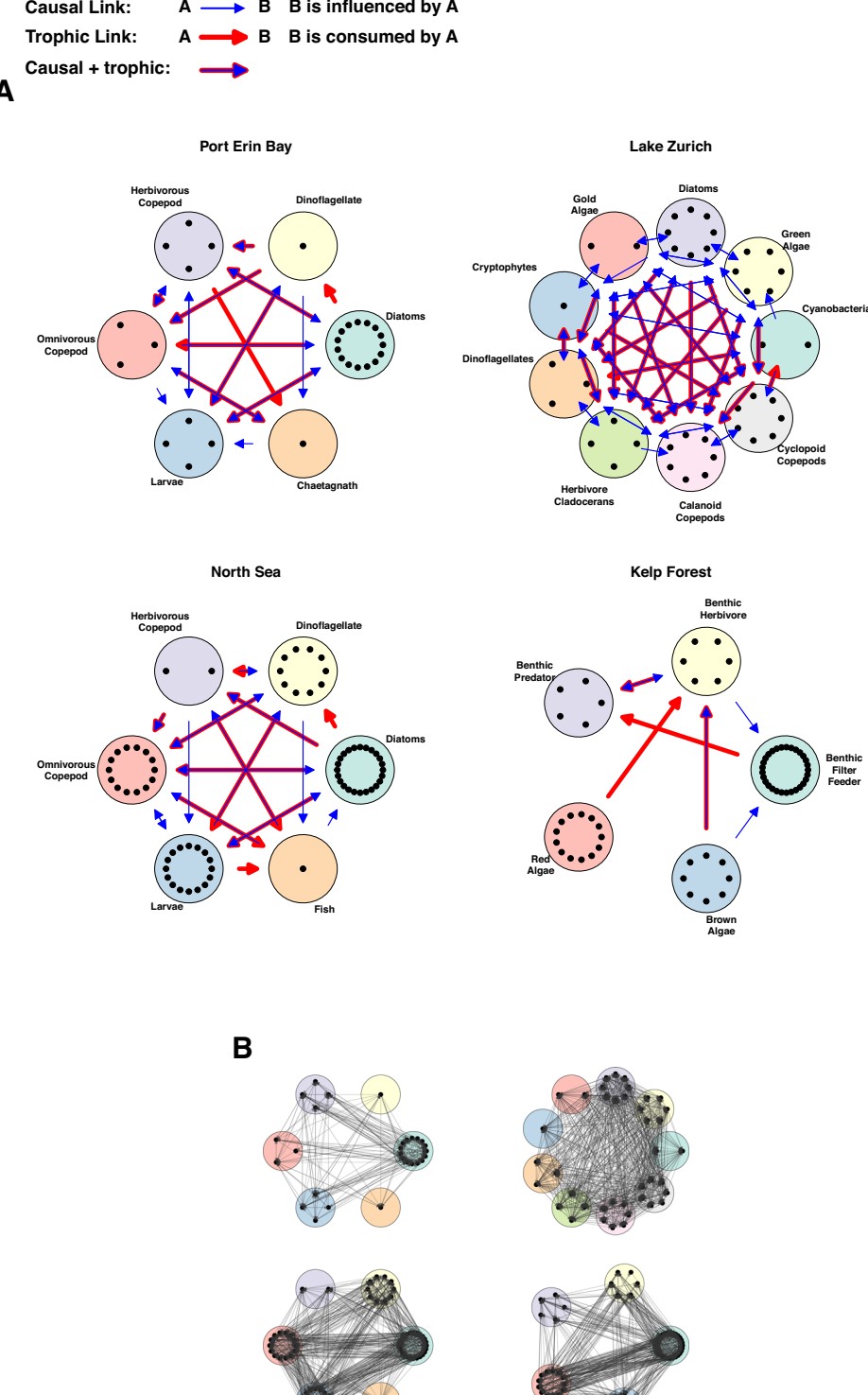

populations, or habitats can reveal the emergent structure of ecosystem networks by smoothing over small scale stochasticity[31–33]. Aggregation can have practical advantages in terms of increased efficiency of data collection, ease of visualization etc. However, others argue that aggregation can also lead to oversimplification that can potentially mask important network properties[34–36].

The model simulations within our study provide a context for the debate on aggregation in ecological research (Fig. 3). In the simulations, timeseries were aggregated to analyze the relationship between model connectance and the aggregate relationship. The model shows that higher connectance at a fine scale translates to higher connectance at a larger aggregate scale: the more links there are connecting individual species across aggregates, the more likely there will be a resolved link between the aggregates in a lower-resolution network. This underscores the sensitivity of CCM to underlying structural parameters of ecological networks and suggests that higher fine-scale connectance, indicative of more densely interconnected ecosystems, can amplify detectable interactions in aggregate data.

We observe a similar association in four real-world systems: higher connectance between individual species across aggregates is associated with an increased likelihood of resolving a significant association between the aggregates. Aggregates are labeled as 'linked' if significant causal connections are detected between them using CCM analysis and 'unlinked' if no such significant connections are found. This distinction highlights how fine-scale interactions can influence the detectability of aggregated causal relationships. However, it is worth noting that the correlation between fine-scale connectance and aggregate interaction strength may not be generalizable to other systems, and is likely sensitive to how timeseries are aggregated and other parametric choices. Further, it is worth noting that not all systems showed statistically significant difference (t-test) when comparing connectance values for the aggregate- and no-aggregate-link groups. However, this may be attributed to the relatively low number of aggregates in each system.

At a high taxonomic resolution, detailed interactions such as specific predator-prey dynamics, competitive interactions, and mutualistic relationships can be revealed, which are crucial for constructing accurate ecological models[26]. These models help in understanding ecosystem dynamics and informing conservation decisions. Conversely, lower taxonomic resolution simplifies ecological data, making it easier to detect broad patterns and general trends, useful for large-scale ecosystem management and identifying key functional groups[29,55].

Temporal resolution also significantly impacts ecological network analysis. High temporal resolution captures rapid interactions and transient dynamics, essential for understanding short-term processes and immediate ecosystem responses. Monthly sampling, for instance, reveals seasonal dynamics and immediate environmental effects. Lower temporal resolution, such as annual sampling, identifies long-term trends and stable relationships that might be obscured by short-term fluctuations[28]. This is beneficial for long-term ecosystem management and planning.

Combining high and low resolutions in both taxonomic and temporal data can provide a more comprehensive understanding of ecosystem dynamics. High-resolution data capture fine-scale interactions and immediate changes, while lower-resolution data highlight broader patterns and long-term stability. This multi-scale approach is crucial for effective ecosystem management and intervention strategies, ensuring that detailed and general trends are both considered in conservation efforts.

In the context of ecosystem management, the absence of detected influence at an aggregate level might overlook vital high-resolution (species-resolved) interactions that are crucial for ecosystem functioning. Such an oversight could lead to management decisions that inadvertently destabilize ecological relationships. This emphasizes the need to consider high-resolution interactions within broader ecosystem management strategies.

Figure 5 shows that while most of the food web links are measurably causal, there are also causal interactions that are non-trophic (blue arrows), and trophic interactions that are non-causal (red arrows). While finding non-trophic causal interactions (blue arrows) is not surprising (competition, mutualism etc.), the converse (known trophic links that measure up as non-causal) is more surprising (red arrows)[56]. At first glance, this looks like a mistake. After all, if a predator consumes a prey, one should expect the predator to have a causal influence on the abundance of the prey. However, we emphasize that lack of a resolved causal link does not indicate a lack of interaction; rather, it reveals there is no resolved influence at that scale at which it was measured. It is likely that the non-causal trophic links may be resolved as causal links when the system is viewed at a higher taxonomic resolution or different timescale.

There have been many advances in tools for inferring causality directly from data. For example, Aurell and Del Ferraro[57] introduced a framework combining correlation-response and dynamic cavity methods, highlighting parallels between Pearl causality and statistical physics. Baldovin, Cecconi, and Vulpiani[23] developed a methodology using the Fluctuation-Dissipation theorem for causal inference, emphasizing correlations and linear response theory. Lastly, recent work by Falasca, Perezhogin, and Zanna[58] demon-

strated how dimensionality reduction and causal inference can reveal significant interactions in high-dimensional systems in the context of climate systems. Future work should explore how ecosystem networks constructed with different methods may change with scale.

## Conclusion

Emerging tools that allow for the collection of higher-resolution ecological data[59] should enable deeper insights into how ecosystems operate. These results show that beyond enabling a fine-scale view, a major advantage of high-resolution data is that it allows viewing the system at multiple time scales. The same principle applies to data collected at high taxonomic resolution. Species-level networks together with networks based on coarser functional aggregates can provide a more robust picture of ecosystem functioning than a high-resolution network can achieve alone, and critically, better reflect emerging quantitative paradigms for ecosystem-based management.

Statements about causality (e.g., "species A influences species B") are usually made in absolute terms without considering the data resolution context. This work can be a reminder that such statements be kept pragmatic, acknowledging that relationships may appear, disappear, or change as systems are viewed at different scales.

## Data availability

Data and code needed to perform the analyses and create the figures are available at https://doi.org/10.6084/m9.figshare.27057952[60].

## Code availability

Data and code needed to perform the analyses and create the figures is available at https://doi.org/10.6084/m9.figshare.27057952[60].

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

## Acknowledgements

This work was supported by the National Science Foundation under grants NSF DEB-1655203 and NSF ABI-1667584, U.S. Department of the Interior's National Park Service under award DOI-NPS-P20AC00527, McQuown Fund and the McQuown Chair in Natural Sciences at the University of California, San Diego. Special thanks to Chase James, Alfredo Giron-Nava, Andrew Johnson, David Dannecker, Bethany Kolody, Adrienne Lee, and Maitreyi Nagarkar initiated the literature-based data gathering. David Johns and Martin Edwards provided expert knowledge for the SAFHOS (North Sea) food web.

## Author contributions

G.S. conceived and directed the study. E.S., T.L., E.D., and G.S. designed the study. E.S. and E.D. cleaned data and performed analyses. E.S., T.L., D.C., E.D., E.M., G.P., J.P., and G.S. drafted the manuscript. E.S. and E.D. generated the figures. All authors provided critical feedback.

## Competing interests

The authors declare no competing interests.
