## [Transparent Peer Review file · Communications Biology]

The Impact of Data Resolution on Dynamic Causal Inference in Multiscale Ecological Networks

Corresponding Author: Dr Erik Saberski

This file contains all reviewer comments in order by version, followed by all author rebuttals in order by version.

Version 0:

Reviewer comments:

Reviewer #2

(Remarks to the Author)
Review attached in the PDF.

Reviewer #3

(Remarks to the Author)

Applying convergent cross mapping (CCM), the authors utilize both modeling data and observational data to explore the impacts of different temporal scales and categorical/functional resolutions on the construction of ecological causal networks. Through simulating the modeling data with individual-based ecological automata models and logistic models, they investigate how dynamics under coarse-grained or fine-grained views across multiple time scales affect relationship detection. Observational data from four aquatic sites, including Port Erin, Lake Zurich, the North Sea, and kelp forests, further validate the exploratory results of the models mentioned. The authors emphasize the importance of data parsing in identifying species interactions, which is well described in the manuscript.

Overall, I believe this paper is a valuable contribution to the literature as detecting relationships from observational data across different time scales is crucial for advancing our understanding of food web structures and further management. Here, I have several suggestions that could improve the manuscript for further submission:

Lines 179 and 199. For CCM, it is well-known that the choice of parameters is crucial for analysis. Here, the direct selection of $E = 4$ and $E = 5$ is quite puzzling, especially since the authors have already pointed out the sensitivity of CCM to embedding dimensions (lines 345-346 and 395). Therefore, either clarify the reasons for choosing these values more clearly or supplement with a thorough robustness test of E to ensure the reproducibility of the results.

Lines 192-194. I apologize if I missed this point, but it is unclear to me why reciprocals are used instead of abundances greater than 1, as this conversion could render species that constitute a significant portion of the population worthless. Indeed, the rationale for capping abundances between 0 and 1 should be detailed so that readers can understand the significance of this step. Furthermore, the definition of parameter C should be explicitly connected.

While keeping results brief is beneficial, some details need to be included. In Figure 2, because observational data can be either taxonomically aggregated or specific to certain species, the taxonomic resolution of the data is unclear. Also, interactions labeled as 'both' are unclear to me. The definition of such interactions and how they are detected should be detailed.

Similar suggestions apply to Figure 4. Only the 'Results' section, line 301, mentions this figure by the authors. However, neither the text nor the figure caption explains the labels 'linked' and 'unlinked,' which can be very confusing to the reader (at least to me).

The discussion thoroughly addresses the importance of taxonomic resolution in resolving causal network structures. However, further discussing these results in terms of species interaction detection (such as food web reconstruction) from observational data, ecosystem management, and intervention strategies might be worthwhile to analyze.

Version 1:

Reviewer comments:

Reviewer #2

(Remarks to the Author)

Dear Editor, I liked the authors' revision and their responses and I am happy to recommend the revised paper for publication.

I would like to stress one final comment on Takens theorem, maybe of help for the authors for future studies. My first, important point on Takens theorem regards the dimensionality of the system. As pointed out by Ruelle (1990), when dealing with high-dimensional systems (effective dimension larger than 7 or 8) we cannot use Takens (it is more than "a challenge"). This is the main limitation, which is somewhat clearer in the revised version. Your study, and the study of Sugihara 2011, deals with intrinsically low-dimensional system: in this case, Takens can be used but larger effective dimension will prohibit its use (even if the theorem is still valid). Regarding stochasticity: thanks for pointing out Stark paper. Even if possible to use a new stochastic Takens theorem, this is not what has been used in the authors paper (and in general I do not see Stark method applied in practice, so I wonder on its contribution to applied nonlinear time series analysis), so it seems to me that the small issue raised by stochasticity seems to remain.

Reviewer #3

(Remarks to the Author)

The authors improved all of those concerns that are mentioned in the previous revision. I think the resubmitted manuscript is ready for future publishing, so I prefer acceptance.

Reviewer #3 (Remarks to the Author):

Applying convergent cross mapping (CCM), the authors utilize both modeling data and observational data to explore the impacts of different temporal scales and categorical/functional resolutions on the construction of ecological causal networks. Through simulating the modeling data with individual-based ecological automata models and logistic models, they investigate how dynamics under coarse-grained or fine-grained views across multiple time scales affect relationship detection. Observational data from four aquatic sites, including Port Erin, Lake Zurich, the North Sea, and kelp forests, further validate the exploratory results of the models mentioned. The authors emphasize the importance of data parsing in identifying species interactions, which is well described in the manuscript.

Overall, I believe this paper is a valuable contribution to the literature as detecting relationships from observational data across different time scales is crucial for advancing our understanding of food web structures and further management. Here, I have several suggestions that could improve the manuscript for further submission:

Lines 179 and 199. For CCM, it is well-known that the choice of parameters is crucial for analysis. Here, the direct selection of $E = 4$ and $E = 5$ is quite puzzling, especially since the authors have already pointed out the sensitivity of CCM to embedding dimensions (lines 345-346 and 395). Therefore, either clarify the reasons for choosing these values more clearly or supplement with a thorough robustness test of E to ensure the reproducibility of the results.

Thank you for pointing this out. We agree with the reviewer that it is important to test the robustness of these results with varying embedding dimensions. To address the reviewer's point, we have included supplemental figures S1 and S3 (attached below). Further, instead of stating fixed embedding dimensions in the manuscript, we clarified this in lines 254-261:

The simulation ran for 15,000 timesteps to generate three distinct abundance time series, one for each trophic category (Fig. 1A). We performed CCM between each time series using $E = 1:10$, $\tau = 1$, $t_p = -1$ (high-resolution web, Fig. 1B middle) and $E=1:10$, $\tau = 500$, $t_p = -500$ (low-resolution web, Fig. 1B right). Only CCM linkages having rho values greater than the maximum absolute cross correlation at any value of E were taken to show nonlinear causal connection. Subtracting out the linear cross-correlation is a simple way to measure whether there are causal dynamics beyond the linear correlation (Deyle *et al.* 2013). Figure S1 shows the CCM results at varying embedding dimensions.

Note that we have also changed the significance test in Figure 1 very slightly to be stricter: instead of considering a link to be significant if the CCM value is greater than the absolute correlation between the lines we have changed it to be greater than the maximum absolute cross correlation between the lines. This is a more rigorous test because it considers potential linear relationships with delays.

None of the above changes the results or main manuscript figures.

Figure S1. Performing CCM in the IBM model with varying embedding dimensions. Red bars signify a non-significant causal interaction (CCM rho < absolute cross correlation). Blue bars signify that a significant relationship was identified. Left charts show CCM computed with tau = 1 and right plots show CCM with tau = 500.

Figure S3. Analogous to Fig. 3, but computed at varying embedding dimensions. Values represent boxplot medians for that embedding dimension.

Lines 192-194. I apologize if I missed this point, but it is unclear to me why reciprocals are used instead of abundances greater than 1, as this conversion could render species that constitute a significant portion of the population worthless. Indeed, the rationale for capping abundances between 0 and 1 should be detailed so that readers can understand the significance of this step. Furthermore, the definition of parameter C should be explicitly connected.

Reciprocals are used here to prevent the model simulation from diverging to infinite values. The reciprocal is only taken at the timestep, in the middle of the simulation, where an absolute value exceeds 1. Thus, this effect will not render an entire time series worthless since it only applies to a fraction of the timesteps.

We added supplemental figure 3 below to demonstrate what the generated timeseries look like from this model. Note that when any timeseries goes outside the bound $[-1, 1]$ it is brought back near 0.

Figure S2. Examples of timeseries generated from the coupled logistic models.

We adjusted lines 270-275 in the text to be:

The abundances were constrained to $[-1,1]$ by taking the reciprocal of any abundance if its absolute value exceeded 1 at time t . Example timeseries can be found in figure S2.

While keeping results brief is beneficial, some details need to be included. In Figure 2, because observational data can be either taxonomically aggregated or specific to certain species, the taxonomic resolution of the data is unclear. Also, interactions labeled as 'both' are unclear to me. The definition of such interactions and how they are detected should be detailed.

Thank you for pointing this out. We have changed lines 388-401 to be more clear:

Similarly, we used CCM to quantify interactions between individual species at both a monthly-scale ($\tau = 1$) and annual-scale ($\tau = 12$) on the three monthly-sampled. This creates two distinct networks for each system: a monthly timescale network and an annual-timescale network. From these networks, interactions between species can be split into three categories: an interaction that appears in *both* networks, an interaction that only appears in the monthly-scale network, or an interaction that only appears in the annual-scale network. Fig. 2 shows the number of network interactions resolved by CCM at each time scale for each of the three categories. All three systems had more interactions resolved at the monthly scale than at the annual scale ($\tau = 12$).

Similar suggestions apply to Figure 4. Only the 'Results' section, line 301, mentions this figure by the authors. However, neither the text nor the figure caption explains the labels 'linked' and 'unlinked,' which can be very confusing to the reader (at least to me).

Thank you for your comments regarding Figure 4. We understand the importance of clearly explaining the labels 'linked' and 'unlinked' to avoid any confusion. We will revise both the text in the Results section and the figure caption to provide a clearer explanation of these terms. Additionally, we will ensure that the paragraph starting at line 573 includes a thorough description.

In the figure caption for Figure 4:

Figure 4. Real world examples showing the relationship between aggregated and fine-scale linkages. Boxplots show that causally linked aggregates (labelled 'linked') have more causal links at the species level (fine-scale), while unlinked aggregates ('unlinked') do not. The center line represents the median, box limits represent upper and lower quartiles, and whiskers are 1.5x interquartile range. P-values are calculated using a one-sided t-test.

In the paragraph starting at line 536:

We observe a similar association in four real-world systems: higher connectance between individual species across aggregates is associated with an increased likelihood of resolving a significant association between the aggregates. Aggregates are labeled as 'linked' if significant causal connections are detected between them using CCM analysis and 'unlinked' if no such significant connections are found. This distinction highlights how fine-scale interactions can influence the detectability of aggregated causal relationships.

The discussion thoroughly addresses the importance of taxonomic resolution in resolving causal network structures. However, further discussing these results in terms of species interaction detection (such as food web reconstruction) from observational data, ecosystem management, and intervention strategies might be worthwhile to analyze.

Thank you for your valuable feedback. We agree that further discussion on the implications of taxonomic resolution in species interaction detection, ecosystem management, and intervention strategies will enhance the manuscript. We have added the following paragraphs to the discussion (lines 555-574) to address these aspects more comprehensively.

At a high taxonomic resolution, detailed interactions such as specific predator-prey dynamics, competitive interactions, and mutualistic relationships can be revealed, which are crucial for constructing accurate ecological models (Dunne, Williams & Martinez 2002). These models help in understanding ecosystem dynamics and informing conservation decisions. Conversely, lower taxonomic resolution simplifies ecological data, making it easier to detect broad patterns and general trends, useful for large-scale ecosystem management and identifying key functional groups (Power *et al.* 1996; Bascompte & Jordano 2007).

Temporal resolution also significantly impacts ecological network analysis. High temporal resolution captures rapid interactions and transient dynamics, essential for

understanding short-term processes and immediate ecosystem responses. Monthly sampling, for instance, reveals seasonal dynamics and immediate environmental effects. Lower temporal resolution, such as annual sampling, identifies long-term trends and stable relationships that might be obscured by short-term fluctuations (Rasmussen *et al.* 2013). This is beneficial for long-term ecosystem management and planning.

Combining high and low resolutions in both taxonomic and temporal data provides a comprehensive understanding of ecosystem dynamics. High-resolution data capture fine-scale interactions and immediate changes, while lower-resolution data highlight broader patterns and long-term stability. This multi-scale approach is crucial for effective ecosystem management and intervention strategies, ensuring that detailed and general trends are both considered in conservation efforts.

The authors focus on the Convergent Cross Mapping (CCM) algorithm as a valuable tool for causal inference from time series. They test the impact of different time scales when inferring causality with CCM. I find the idea of testing the contribution of different time scales in the context of CCM interesting and important and I like the use of dynamical systems tools to infer causality.

I find however the manuscript difficult to follow, especially in the methods and results section. Because of this, the current version should undergo a major revision. In addition, I would also like to stress few points regarding Takens' embedding theorem which may be important to emphasize, especially when mentioning the limitation of the CCM method.

In what follows I will start with a few comments on the Takens embedding theorem, I will then focus on what I find confusing in the manuscript.

Important limitation of considering Takens' embedding theorem.

- The skill of Takens embedding theorem depends on finding good analogues (i.e., https://journals.ametsoc.org/view/journals/atsc/26/4/1520-0469_1969_26_636_aparbn_2_0_co_2.xml) for the process we care about. It can be shown that even in deterministic system this is not possible for high-dimensional systems. It is therefore important to state that the CCM method would work for systems that are intrinsically low-dimensional. A good discussion on the topic can be found in this paper: <https://www.ncbi.nlm.nih.gov/pmc/articles/PMC7512371/> ; page 11/29 (also citations therein). In general time series reconstruction in the case of high dimensional systems is not possible and it should be stated clearly. Another useful reference on the topic is given by Ruelle: <https://royalsocietypublishing.org/doi/10.1098/rspa.1990.0010>.

Let me emphasize the point I want to make: this does not mean that the CCM method is not useful, but that it works on systems that are intrinsically low dimensional. One "solution" is to focus on systems where there is a clear distinction between short and long time scales and then focus on the longer time scales only; with the assumption that the "new" system spanned by the slow temporal dynamics would also be lower dimensional.

On the other hand, this also means that, at least for high-dimensional systems, focusing on different time scales in the embedding process may show inherently bad results when considering the shorter time scales for the reasons stated above.

- Takens' theorem is not valid for stochastic dynamics. So, applications of the method assume that time series come from a deterministic process. In the case of real-world measurements this may also be a limitation as each time series measurements will be influenced by background noise. This should be also stated as a possible limitation or at least briefly discussed.

We appreciate the reviewer's concern regarding the application of Takens' theorem to stochastic dynamics and the potential limitations arising from background noise in real-world measurements. We acknowledge that Takens' original theorem assumes a deterministic system, which may indeed pose a limitation when dealing with stochastic processes.

However, Takens' theorem has been extended to accommodate certain classes of stochastic dynamical systems, as demonstrated by Stark et al. (Stark *et al.* 1997). These generalizations allow for the reconstruction of system dynamics even in the presence of stochasticity. Stark et al. showed that Takens' embedding theorem could be extended to forced and stochastic systems, ensuring that our methods remain valid and robust despite the presence of noise. Additionally, the simplex and S-map algorithms we employed are designed to handle observation noise effectively, relying on low-dimensional deterministic relationships within the system. A great example of this is demonstrated in (Sugihara *et al.* 2011)

Thus, while acknowledging the reviewer's point, we assert that our methods remain applicable and robust for the analysis presented in our study. We will include a brief discussion in the revised manuscript to address this potential limitation and clarify the robustness of our approach.

We have added the following paragraph in lines 116-128:

Convergent Cross-Mapping (CCM) has its limitations, particularly in the context of high-dimensional systems where finding good analogues, as required by Takens' Theorem, can be challenging. Work by Bradley and Kantz (Bradley & Kantz 2015), Ruelle (Ruelle 1990), and Baldovin, Cecconi, and Vulpiani (Baldovin, Cecconi & Vulpiani 2020) highlights the difficulties of time series reconstruction in such high-dimensional contexts. Although Takens' theorem was originally formulated for deterministic systems, extensions by Stark et al. (Stark *et al.* 1997) have expanded its applicability to certain classes of stochastic dynamical systems. Further, the simplex and S-map algorithms utilized in our empirical dynamic modeling (EDM) analyses are specifically designed to handle observation noise, relying on the existence of low-dimensional deterministic relationships between variables. A good example of this is shown in (Sugihara *et al.* 2011). These extensions ensure that our embedding procedures remain valid and robust, even in the presence of stochastic elements, thus supporting the applicability of CCM to real-world measurements.

Line 104-106. "None of these methods are both...require all relevant causal variables to be observed".

- I agree on this. On the other hand, as shown above, the embedding theorem also comes with few cons. Such cons should be acknowledged. In general, a useful solution to the fact that "all relevant causal variables..." comes from stochasticity. Specifically, by focusing on the "right time scales", it is possible to focus on few "proper" variables (the

ones with longer timescales) and parametrize the rest as noise. This is useful to consider when using the method of Runge and it can be briefly stated in the introduction.

We agree that while the embedding theorem provides powerful tools for causal inference, it also has limitations that need to be acknowledged. We hope the addition in lines 116-128 (above) explain such limitations.

We have also added the following sentence in line 108-110 if the introduction, as suggested by the reviewer:

It is worth noting that when it is not possible to measure all system variables, focusing on the dominant variables and treating the others as noise can be beneficial

Lines 95-97. On interventions.

- A new set of tools to infer causality from data in the “interventional way” comes from methods stemming from statistical physics (e.g., linear response theory, or better Fluctuation-Response formalism). Some of these tools can be added to the discussion or considered for future studies. The three papers showing methodological advancements in this direction are mainly:

A. The paper of Aurell and Del Ferraro who first recognized the parallelism between Pearl causality and tools from statistical Physics. <https://iopscience.iop.org/article/10.1088/1742-6596/699/1/012002>

B. The paper of Baldovin, Cecconi and Vulpiani who first proposed a practical methodology coming from Fluctuation-Dissipation theorem to tackle causal inference <https://journals.aps.org/prresearch/abstract/10.1103/PhysRevResearch.2.043436>

C. The paper of Falasca, Perezhogin, Zanna using these tools for causal inference in climate data <https://journals.aps.org/pre/abstract/10.1103/PhysRevE.109.044202>

Thank you for these suggestions. We have added the following paragraph to the end of the discussion offering an avenue for potential future research.

There have been many advances in tools for inferring causality directly from data. For example, Aurell and Del Ferraro (2016) introduced a framework combining correlation-response and dynamic cavity methods, highlighting parallels between Pearl causality and statistical physics. Baldovin, Cecconi, and Vulpiani (2020) developed a methodology using the Fluctuation-Dissipation theorem for causal inference, emphasizing correlations and linear response theory. Lastly, recent work by Falasca, Perezhogin, and Zanna (2024) demonstrated how dimensionality reduction and causal inference can reveal significant interactions in high-dimensional systems in the context of climate systems. Future work should explore how ecosystem networks constructed with different methods may change with scale.

Line 146: “Because nonlinearity implies”

- True. However, focusing on different time scales often translates into a different effective dynamics of the system. In other words, similar to the points made before: by focusing on different time scales we are also focusing on different types of dynamics, often linearity can play an important role for the slower time scale.

We agree and believe that our study clearly emphasizes the importance of examining different time scales, which can indeed reveal various effective dynamics within the system. We hope this is evident in our discussion, particularly regarding how different time scales may highlight different types of dynamics.

Methods section.

- I would recommend starting the method section with a mathematically clear review of the CCM method. This is important in my opinion as the “1) Model and Field Data” section depends on it.

Thank you for the comment. We have added a section that details the CCM method in the beginning of the methods section.

Regarding the automata model: is there any citation to this model? Or is it a novel model tested in this work? In the first case, add citation.

This is a novel model specifically designed to isolate and illustrate the interactions occurring at multiple time scales, which we then apply to empirical (observational) time series in our study.

Section “Aggregation in a logistic model”. It would be of great help writing down the equations and/or pointing out to the right literature; saying “a simple logistic model” is not enough. Equations will help to better understand the discussion from Line 186 to Line 195, which right now is not clear.

Thank you for pointing this out. We have added equations to this part of the methods section.

The metrics “Fine-Scale Connectance”, “Resolved Aggregate Interaction Strength” and “Aggregated Functional Group Linkage” are briefly mentioned in the Method section. However, it would be useful to add a subsection in the method section titled “metrics” which clearly explains these two quantities. They are important in the context of Figure 3 but are really only briefly mentioned in the manuscript.

We have added a metrics subsection to the beginning of the methods section.

- Figure 1:

Please add time [t] as a label of the x-axis for each time series.

Thank you for pointing this out – the label has been added.

I am confused here and cannot understand what the "Food Web" network stand for. Can this be considered as a ground truth? If not please explain. Apologies if this explanation is in the results section and I missed it.

The "Food Web" network in Figure 1 represents the interactions among different trophic levels within the simulated ecological system. It is constructed based on the known (programmed) trophic interactions and is used as a reference to compare against dynamically inferred causal networks.

The food web is not considered "ground truth" in the strictest sense but serves as a foundational reference for what the direct interactions are in this network. The goal is to assess how well the dynamically inferred causal relationships (using Convergent Cross Mapping, CCM) correspond to these known trophic interactions. There are more causal relationships in the network than food web linkages because the food web does not show indirect interactions (e.g., the effect of resources on secondary consumers).

It is useful if when referring to the different panels in Figure 1, such panels are also labelled; e.g. panel (a), (b) etc. referring to the 3 time series or the graphs.

Thank you. We have added the labels.

In general, I would recommend rewriting the discussion of Figure 1. As it is right now it is not clear what it should be the ground truth. It is also not clear what does the bidirectionality means: for example, in the 1-timestep graph, I understand the link $R \rightarrow PC$, but why should we also have $PC \rightarrow R$? This is much less clear to me.

Thank you for pointing this out. We have added the following to the beginning of the discussion to make it clear what the expected interactions should be:

The individual-based automata (IBA) is a great example of interactions occurring at varying timescales in a controlled system. Here, we can map out all expected interactions as a ground truth:

Direct interactions:

- **Primary Consumers (PC) \rightarrow Resources (R):** Primary consumers influence resource abundance by consuming them. This interaction occurs at a **1-timestep scale** (PC \rightarrow R).
- **Resources (R) \rightarrow Primary Consumers (PC):** This bi-directional relationship exists because the survival and reproduction of primary consumers depend on the availability of resources. This interaction also occurs at a **1-timestep scale** (R \rightarrow PC).
- **Secondary Consumers (SC) \rightarrow Primary Consumers (PC):** Secondary consumers prey on primary consumers, lowering the PC their abundance. This direct interaction occurs at a **1-timestep scale** (SC \rightarrow PC).

- **Primary Consumers (PC) -> Secondary Consumers (SC):** The survival and reproduction of secondary consumers depends on the availability of primary consumers. This direct interaction occurs at a **500-timestep scale (PC -> SC)**.

Indirect interactions:

- **Resources (R) -> Secondary Consumers (SC):** Indirectly, the availability of resources influences secondary consumers by affecting the population of primary consumers, which are prey for secondary consumers. This indirect interaction is resolved at about a **500-timestep scale (R -> PC-> SC)**.
- **Secondary Consumers (SC) -> Resources (R):** Indirectly, secondary consumers influence resource abundance by affecting the population of primary consumers, which consume the resources. This indirect interaction is also resolved at a **1-timestep scale (SC -> PC-> R)**.

Thus, there are a total of six expected interactions between three species, which implies full connectance (everything causally influences everything). However, these interactions span different timescales.

- Baldovin, M., Cecconi, F. & Vulpiani, A. (2020) Understanding causation via correlations and linear response theory. *Physical Review Research*, **2**, 043436.
- Bascompte, J. & Jordano, P. (2007) Plant-animal mutualistic networks: the architecture of biodiversity. *Annual review of ecology, evolution, and systematics*, 567-593.
- Bradley, E. & Kantz, H. (2015) Nonlinear time-series analysis revisited. *Chaos: An Interdisciplinary Journal of Nonlinear Science*, **25**.
- Deyle, E.R., Fogarty, M., Hsieh, C.-h., Kaufman, L., MacCall, A.D., Munch, S.B., Perretti, C.T., Ye, H. & Sugihara, G. (2013) Predicting climate effects on Pacific sardine. *Proceedings of the National Academy of Sciences*, **110**, 6430-6435.
- Dunne, J.A., Williams, R.J. & Martinez, N.D. (2002) Food-web structure and network theory: the role of connectance and size. *Proceedings of the National Academy of Sciences*, **99**, 12917-12922.
- Power, M.E., Tilman, D., Estes, J.A., Menge, B.A., Bond, W.J., Mills, L.S., Daily, G., Castilla, J.C., Lubchenco, J. & Paine, R.T. (1996) Challenges in the quest for keystones: identifying keystone species is difficult—but essential to understanding how loss of species will affect ecosystems. *BioScience*, **46**, 609-620.
- Rasmussen, C., Dupont, Y.L., Mosbacher, J.B., Trøjelsgaard, K. & Olesen, J.M. (2013) Strong impact of temporal resolution on the structure of an ecological network. *PLoS one*, **8**, e81694.
- Ruelle, D. (1990) The Claude Bernard Lecture, 1989-Deterministic chaos: the science and the fiction. *Proceedings of the Royal Society of London. A. Mathematical and Physical Sciences*, **427**, 241-248.

Stark, J., Broomhead, D.S., Davies, M.E. & Huke, J. (1997) Takens embedding theorems for forced and stochastic systems. *Nonlinear Analysis: Theory, Methods & Applications*, **30**, 5303-5314.

Sugihara, G., Beddington, J., Hsieh, C.-h., Deyle, E., Fogarty, M., Glaser, S.M., Hewitt, R., Hollowed, A., May, R.M. & Munch, S.B. (2011) Are exploited fish populations stable? *Proceedings of the National Academy of Sciences*, **108**, E1224-E1225.

Reviewer #2 (Remarks to the Author):

Dear Editor, I liked the authors' revision and their responses and I am happy to recommend the revised paper for publication.

I would like to stress one final comment on Takens theorem, maybe of help for the authors for future studies. My first, important point on Takens theorem regards the dimensionality of the system. As pointed out by Ruelle (1990), when dealing with high-dimensional systems (effective dimension larger than 7 or 8) we cannot use Takens (it is more than "a challenge"). This is the main limitation, which is somewhat clearer in the revised version. Your study, and the study of Sugihara 2011, deals with intrinsically low-dimensional system: in this case, Takens can be used but larger effective dimension will prohibit its use (even if the theorem is still valid). Regarding stochasticity: thanks for pointing out Stark paper. Even if possible to use a new stochastic Takens theorem, this is not what has been used in the authors paper (and in general I do not see Stark method applied in practice, so I wonder on its contribution to applied nonlinear time series analysis), so it seems to me that the small issue raised by stochasticity seems to remain.

Thank you for your constructive feedback and for recommending our revised paper for publication.

Regarding your comment on the use of Takens' theorem, we appreciate suggesting the added clarification on its limitations, particularly in high-dimensional systems as highlighted by Ruelle (1990). However, for the systems we explored, we believe its application remains appropriate. We also acknowledge that in future studies involving higher-dimensional systems, alternative approaches may be required.

To add, in the Simplex prediction algorithm used in Convergent Cross Mapping (CCM) within our paper, the embedding dimension is closely tied to other parameters such as the number of neighbors used for prediction and the timescale of the system's dynamics. Thus, it is not uncommon to observe relatively high embedding dimensions (e.g., $10 < E < 20$) performing well in the presence of noise or when the system's dominant dynamics operate over longer timescales than the data sampling interval. Further exploration of how embedding dimension interacts with these parameters could be a valuable direction for future research.

Reviewer #3 (Remarks to the Author):

The authors improved all of those concerns that are mentioned in the previous revision. I think the resubmitted manuscript is ready for future publishing, so I prefer acceptance.

Thank you! We greatly appreciate your feedback.